# Towards Understanding the Direct and Indirect Actions of Growth Hormone in Controlling Hepatocyte Carbohydrate and Lipid Metabolism

**DOI:** 10.3390/cells10102532

**Published:** 2021-09-24

**Authors:** Mari C. Vázquez-Borrego, Mercedes del Rio-Moreno, Rhonda D. Kineman

**Affiliations:** Department of Medicine, Section of Endocrinology, Diabetes, and Metabolism, University of Illinois at Chicago and Research and Development Division, Jesse Brown VA Medical Center, Chicago, IL 60612, USA; mavazbor@uic.edu (M.C.V.-B.); mdelri3@uic.edu (M.d.R.-M.)

**Keywords:** growth hormone, hepatocyte, metabolism, IGF1, insulin, carbohydrates, lipids

## Abstract

Growth hormone (GH) is critical for achieving normal structural growth. In addition, GH plays an important role in regulating metabolic function. GH acts through its GH receptor (GHR) to modulate the production and function of insulin-like growth factor 1 (IGF1) and insulin. GH, IGF1, and insulin act on multiple tissues to coordinate metabolic control in a context-specific manner. This review will specifically focus on our current understanding of the direct and indirect actions of GH to control liver (hepatocyte) carbohydrate and lipid metabolism in the context of normal fasting (sleep) and feeding (wake) cycles and in response to prolonged nutrient deprivation and excess. Caveats and challenges related to the model systems used and areas that require further investigation towards a clearer understanding of the role GH plays in metabolic health and disease are discussed.

## 1. Introduction

Growth hormone (GH) plays an important role in modulating metabolic function throughout the lifespan. This review builds on excellent preceding reviews on this topic [1,2] and will specifically focus on the model systems used, including caveats and challenges, to inform our current understanding of the molecular mechanisms by which GH regulates liver (hepatocyte) carbohydrate and lipid metabolism. To date, there has been evidence that GH controls hepatocyte metabolism indirectly by controlling substrate availability to the liver. This action is in large part mediated by the GH/GH receptor (GHR) regulation of lipid mobilization from white adipose tissue (WAT), as well as via GH-mediated alterations in insulin-like growth factor I (IGF1) and insulin production and function. In addition, evidence that GH also acts directly on hepatocytes through the GHR to alter substrate metabolism is accumulating. These actions of GH are critical to maintain nutrient balance under normal physiologic conditions, where alterations in the GH production or function can contribute to the progression of metabolic disease.

## 2. Caveats and Challenges Relevant to the Study of the GH-Mediated Regulation of (Hepatocyte) Metabolism

### 2.1. Metabolic Actions of GH/GHR Can Be Mediated by Multiple Intracellular Signaling Pathways

The actions of GH are mediated through the GHR, a type I cytokine receptor. Al-though the most studied action of GH/GHR is via the sequential phosphorylation (activation) of Janus kinase 2 (JAK2) and the signal transducer and activator of transcription 5 (STAT5), it is important to keep in mind that GHR/JAK2 also activates STAT1/STAT3, RAS/RAF/ERK1/2, and PI3K/Akt/mTOR [3,4]. Independent of JAK2, GHR can activate the Src family kinase (SFK) signaling, leading to ERK1/2 activation via PLCγ and PKC signaling [3]. It should be noted that the (patho)physiologic impact of these STAT5-independent actions of GHR are just beginning to be explored. In addition, evidence showing that GHR can localize to the nucleus and modulate gene expression is accumulating [5].

### 2.2. Limitations of In Vitro Models and Choice of Ligand Used

In this review, the terms “liver” and “hepatic” are used to refer to whole organ extracts and function, where it is important to keep in mind that the healthy liver is made of ~70% parenchymal cells (hepatocytes) while the remaining cells are represented by cholangiocytes, hepatic-stellate cells, vascular endothelial cells, and resident macrophages (Kupffer cells) and infiltrating immune cells. Therefore, the most direct approach to studying how GH/GHR regulates hepatocyte metabolism is the use of primary hepatocytes or hepatocyte cell lines. However, primary hepatocytes rapidly dedifferentiate in culture [6,7] with a decline in GHR expression and activation [8]. In addition, hepatocyte cell lines (many originating from hepatomas) do not express the GHR, or expression is clone-specific, requiring stable transfection of the GHR to study the actions of GH on cellular metabolism. Current efforts to develop “organs (liver) on a chip” are underway to sustain differentiated function, where these systems may increase the translatability of in vitro studies [9]. It should also be noted that in many studies, species heterologous GH is used, where subtle differences in the ligand used to activate the GHR may alter the ultimate impact on downstream signals [10,11]. In addition, human GH (hGH), initially favored due to its availability in a recombinant form, has been used to test metabolic endpoints in rodent models in vivo and in vitro. However, hGH binds and activates both the GHR and the prolactin receptor (PRLR) in rodents [12], making it difficult to determine which changes are exclusively GHR-dependent, particularly in vivo, where prolactin has been shown to have profound effects on metabolic function [13]. However, the short form of the PRLR, which cannot activate JAK2/STAT5, is the dominant form expressed in the mammalian liver [14,15]. Therefore, the in vitro use of hGH in rodent hepatocyte cultures may still provide clues as to the contribution of GHR/STAT5 in mediating metabolic endpoints [16].

### 2.3. Challenges Related to In Vivo Approaches (Species, Exogenous/Endogenous Pattern of GH Delivery, Sex, and Metabolic Context)

As indicated above, and detailed in subsequent sections, changes in GH levels and signaling can alter insulin production and function. Therefore, to help isolate the specific role of GH from insulin in regulating metabolic processes in vivo, human studies utilize hyperinsulinemic/euglycemic clamp techniques to maintain insulin and glucose in a well-defined range while varying GH input. Under these conditions, the incorporation of stable isotope tracers into metabolites provides clues as to how GH regulates substrate metabolism in different physiologic and disease states (literature reviewed in [1,17,18]). Although a wealth of information has been derived from this strategy, particularly related to the actions of GH on adipose and skeletal muscle metabolism, it remains difficult to tease apart the direct actions of GH on hepatocyte metabolism from its actions on extrahepatic tissues, because liver biopsies from subjects without clinically relevant features of liver disease are not warranted. Therefore, in vivo animal models have been used to delve deeper into the mechanisms by which GH regulates hepatocyte metabolism. Many of the early studies using animal models focused on agricultural species, due to the potential to enhance growth, while rats were studied in the laboratory. In these larger models, it is possible to collect serial blood samples to assess pulsatile patterns of GH release, which is critical for its ultimate function. However, mice have become the favored model system due to the ease of genetic manipulation, but serial blood sampling remains a challenge due to a low blood volume and is only performed in selected laboratories [19,20,21,22]. In rats, the pattern of pulsatile GH release is sexually dimorphic, with males exhibiting regular high-amplitude pulses and low nadirs, while females release GH in a more continuous pattern [23,24]. A similar pattern was also observed in mice [19,25]. In humans, GH release is also sex-dependent, with mean GH levels higher in premenopausal females compared to those of males [26]. In both rats and mice, the hepatic expression of a large set of genes have been shown to be sexually dimorphic and dependent on the pattern of GH delivery. In fact, the continuous delivery of exogenous or endogenous GH to males shifts the pattern of hepatic expression to that observed in females [26,27]. These sex-dependent differences in gene expression are due, in part, to differential actions of the GH pattern on STAT5 DNA binding to gene promoters, as well as STAT5-dependent alteration in DNA methylation patterns [24,28,29]. In addition, estrogens regulate hepatic gene expression (including IGF1) dependent and independent of alterations in GH release [30,31]. Therefore, with respect to understanding the role GH plays on liver metabolism, caution should be used when extrapolating findings between males and females.

GH levels are reduced in the context of diet-induced obesity in humans, rats, and mice [26,32,33,34]. In states of nutrient deprivation (fasting, caloric restriction, and type I diabetes), GH levels are increased in humans, as well as mice [35,36,37,38,39,40]. However, in mice, the fasting-induced rise in GH is dependent on the duration/severity of nutrient deprivation and methods of sampling [21,41,42]. In contrast, pulsatile GH release is suppressed in nutrient-deprived male rats [43]. When considering the ultimate contribution of these (patho)physiological changes in GH to metabolic function, it is important to keep in mind major differences in metabolic rate between humans and rodent models. Under standard ambient temperatures (22 °C), adult humans expend minimal energy beyond the basal metabolic rate (BMR) to maintain core body temperature (i.e., the thermal neutral zone), in part related to their low surface-to-body mass ratio. Therefore, humans lose <2% of body weight after a 4-day fast, when activity level is controlled. However, since rats and mice have a high surface-to-body mass ratio, when housed under standard temperatures (22 °C), significant energy is required beyond the BMR to sustain core body temperature [44,45,46,47]. Therefore, laboratory rodents rapidly lose body weight, when food is restricted. Mice lose up to 10% of their starting body weight after an overnight fast (the time when >75% of calories are normally consumed), with a ~20% weight loss after a 48 h fast [44]. Therefore, even a short duration fast in rodents represents an extreme catabolic state, where the modulatory actions of GH on metabolic function may be masked by compensatory mechanisms to survive. This is particularly relevant in fasted mice that undergo torpor (reduction in core body temperature) within 24 h of total food withdrawal, which is associated with reduced brain and motor activity and glucose levels [45]. The high metabolic rate in rats and mice is also a challenge when examining the contribution of GH to metabolic function under conditions of nutrient excess [46,47]. Diet-induced obesity and insulin resistance are slow to develop in rodents, and the transition to severe metabolic disease such as diabetes type II, hyperlipidemia, cardiovascular disease, and non-alcoholic steatohepatitis (NASH) does not occur unless additional environmental or genetic alterations are used. Given these challenges, many researchers interested in studying metabolic function using mouse models have transitioned to housing within the thermal neutral zone (28–30 °C). A few groups focusing on the GH-axis function have implemented this strategy [48,49,50], but none, to date, have specifically studied the impact on the GH-mediated regulation of hepatic metabolism under thermoneutral conditions.

### 2.4. Power and Limitations of Genetically Engineered Mouse Models

Despite the notable differences in rodent vs. human metabolism, the use of genetically engineered mouse models has dramatically advanced our understanding of the role GH plays in regulating metabolic function. Multiple models have been generated with the congenital whole-body knockout or expression of transgenes that impact GH production and signaling [51,52]. Congenital models with the knockout of GH, GHR, or STAT5 or those that express a GHR-antagonist are dwarf with an increase in the ratio of fat to lean mass, while models with the overexpression of species heterologous GH transgenes are giant, with a decrease in the ratio of fat to lean mass. Although models with congenital genetic modifications have been extremely informative regarding the integrative physiology of early-onset GH deficiency (GHD), GH insensitivity, and GH excess, it is difficult to determine if the liver-specific changes observed are due to the direct actions of GH on hepatocyte function, changes in body size and composition, or other secondary changes owing to a lifetime of alterations in IGF1 or insulin production (for details, refer to the following section). To learn more regarding the tissue-specific and age-dependent actions of GH, investigators have utilized the Cre recombinase (Cre)/loxP system to knockout the GHR and downstream signals in a tissue-specific fashion [53,54]. In the majority of studies that specifically examine the impact of disrupting hepatocyte GH-signaling, GHR-floxed mice are crossbred with mice expressing Cre recombinase driven by promoters expressed early in hepatocyte development. Once again, early changes in GH, IGF1, and insulin impact the development of other metabolically relevant tissues, confounding the interpretation of the liver phenotype. To circumvent this problem and investigate metabolic dysfunction that occurs more frequently in adults, tissue-specific and inducible models have been developed. Of note, a mouse model of whole body inducible GHR knockdown has been recently developed to examine the age-dependent impact of reducing GHR signaling on metabolic function and lifespan [55,56,57]. To drive adult-onset, hepatocyte-specific genetic manipulation, researchers have turned to the use of an adeno-associated viral (AAV) vector, serotype 8 that has a high tropism to the liver. This vector is used to express Cre recombinase, as well as other transgenes, under a hepatocyte-specific promoter (thyroxine-binding globulin (TBG)). Relative to the use of adenoviral vectors, AAV vectors only evoke a mild immune reaction, have minimal off-target effects and induce efficient and persistent transgene expression [58]. To date, this strategy has been specifically used in mice to study the impact of adult-onset, hepatocyte-specific GHR knockdown (aHepGHRkd), without or with the hepatocyte IGF1 reconstitution or the expression of constitutively active STAT5b [59,60,61,62]. Limitations of this system include the fact that the AAV vector does not consistently incorporate into the genome, so if it is applied to rapidly dividing cells or the transgene induction enhances cell division, the vector may become diluted in subsequent cell generations. However, it is important to take into account that chromosomal integrations can randomly occur in the host genome [63]. In addition, antibodies do develop to AAVs, precluding the efficient expression of the transgene with a sequential application of AAV vectors.

## 3. Understanding the Interconnection between GH, IGF1, and Insulin Is Critical to Understand How GH Impacts Hepatocyte Metabolism

### 3.1. Tissue-Specific Metabolic Effects of Insulin and IGF1—Common and Divergent

As recently reviewed [64], insulin and IGF1 regulate growth and metabolism through their cognate tyrosine kinase receptors (insulin receptor (INSR) and IGF1 receptor (IGF1R), respectively). In brief, in skeletal muscle, insulin promotes glucose uptake/utilization and glycogen synthesis and protects against protein breakdown. In WAT, insulin promotes glucose uptake and lipogenesis and blocks lipolysis. In the liver, glucose uptake is not insulin-dependent; however, insulin inhibits glucose production, lipid oxidation, and very low-density lipoprotein (VLDL) release and stimulates glycogen synthesis and lipogenesis. The IGF1R shares high homology to the INSR, and there is evidence that the IGF1R and the INSR can form hybrid receptors which have higher affinity to IGF1 [65]. In general, IGF1/IGF1R signaling plays a critical role in cell growth during development, while IGF1 also has insulin-like effects that are dependent on the relative level of IGF1R expression in target tissues [66,67,68]. IGF1 promotes adipocyte development, but the expression of the IGF1R is low in mature adipocyte, where insulin’s actions dominate to enhance glucose uptake, stimulate lipogenesis and inhibit lipolysis [69,70,71]. In skeletal muscle, IGF1 is critical for muscle development [72] but is dispensable in the control of glucose metabolism in mature myocyte [73]. In contrast, the mature hepatocyte does not express the IGF1R [74].

### 3.2. IGF1 and Insulin Regulate GH Production and Vice Versa

As reviewed below, GH regulates the production and action of both IGF1 and insulin, which in turn impacts GH production and function. Therefore, to understand the role GH plays in regulating hepatocyte metabolism in health and disease, it is also important to understand the interconnection between GH, IGF1, and insulin (Figure 1). GH is produced by somatotropes of the anterior pituitary gland. The primary control of GH synthesis and release is governed by the hypothalamic neuropeptides, GH-releasing hormone (GHRH, positive), and somatostatin (SST, negative). GH is released into the general circulation in a sex-dependent, pulsatile pattern that is species-dependent [19,23,24,25,26]. The GHR is expressed in a wide variety of tissues [75]. Although IGF1 is produced by multiple tissues [76], the hepatocyte is the major contributor to circulating IGF1 [77]. GH acts directly on the hepatocyte via GHR signaling through JAK2/STAT5 to stimulate *Igf1* gene expression. GHR/JAK2/STAT5 also stimulates the expression of IGF acid labile subunit (*Igfals*; [78]). IGFALS and IGF1-binding protein 3 (IGFBP3) form a ternary complex with IGF1 to maintain IGF1 in the circulation and modulate the IGF1 availability at target tissues [68,79]. GH negatively feeds back at the level of the hypothalamus through the GHR to reduce GH secretion [80]. Although historical evidence suggests that IGF1 does not play a major role in the central regulation of GH secretion [81,82,83,84], more recent studies examining the tissue-specific loss of the IGF1R on GHRH neurons support a central role [85]. In addition, IGF1 acts through the IGF1R on pituitary somatotropes to suppress GH synthesis and release [86,87,88]. This negative feedback system serves to maintain circulating GH levels in a well-defined range. GH and IGF1 have been shown to alter insulin production associated with changes in pancreatic β-cell mass and/or function [89,90,91,92,93,94]. Pancreatic β-cell-specific knockout of the GHR [90] or the IGF1R [95,96] indicate GH and IGF1 directly support optimal insulin synthesis and release, where the GHR and the IGF1R may physically interact to further augment β-cell proliferation [97]. In turn, insulin supports hepatic IGF1 production by direct stimulation of *Igf1* gene expression, as well as sustaining hepatic *Ghr* expression [98,99]. To counterbalance this system, insulin acts through INSRs on pituitary somatotropes to suppress GH secretion [87,100]. Insulin may also act centrally to regulate GH release, but the mechanisms involved remain to be defined [101].

### 3.3. GH Antagonizes the Actions of Insulin in a Tissue- and Context-Specific Manner

It is commonly accepted that GH antagonizes the actions of insulin, where chronic GH excess is associated with hyperinsulinemia, with or without hyperglycemia. However, it is appreciated the antagonistic action of GH on insulin’s actions is dependent on the relative balance between GH, IGF1, and insulin, nutritional status, and the specific tissue examined [102]. In animal models, GH is thought to antagonize actions of insulin (reducing glucose uptake and inhibiting lipid accumulation) in adipocytes and skeletal muscle by promoting p85 accumulation, which in turn sequesters insulin receptor substrate (IRS), thereby impairing insulin-mediated PI3K/Akt activation (for review, see [102]). However, this GH-mediated impairment in canonical insulin signaling has not been observed in humans. The anti-insulin effects of GH in adipocytes may be mediated by STAT5, since the adipocyte-specific loss of either GHR [103] or STAT5 [50,104] leads to an increase in adipose tissue mass under chow-fed conditions and improves whole body insulin sensitivity. However, in contrast to the adipocyte-specific loss of GHR, the loss of adipocyte STAT5 is not associated with a reduction in WAT lipolysis and does not impact the ability of GH treatment to reduce WAT mass [50]. These results suggest that GH/GHR signaling acts independently of STAT5 to control WAT mass. Consistent with these observations, in human adipocytes, GH inhibits lipogenesis and stimulates lipolysis by the GHR/JAK2-mediated activation of ERK, which in turn reduces PPARγ/FSP27 activity and stimulates hormone sensitive lipase (HSL) activity [105,106,107]. It is becoming appreciated, at least in humans, that this lipolytic actions of GH indirectly contributes to impaired skeletal muscle glucose utilization, independent of alterations in canonical insulin signaling, via substrate competition with non-esterified fatty acids (NEFA) [108,109]. Consistent with these actions of GH in WAT and skeletal muscle, mice with congenital GHD due to GHRH knockout (GHRH^−/−^, [110]), mice expressing a mutant GH receptor that cannot activate STAT5 (GHR-391; [111]) and mice with adult-onset isolated GHD due to the selective destruction of the pituitary somatotropes (AOiGHD; [112]) exhibit enhanced glucose uptake in skeletal muscle and adipose tissue under hyperinsulinemic/euglycemic clamp conditions, while mice with elevated GH levels due to the congenital loss of hepatocyte IGF1 (liver IGF1 deficiency (LID); [113,114]) or the deletion of the IGF1R and INSRs in somatotropes (HiGH; [112]) exhibit impaired insulin-mediated glucose uptake.

The ultimate impact of altered GH production and signaling on hepatic insulin sensitivity is not as clear-cut. The insulin-mediated suppression of hepatic glucose production (HGP), as assessed under hyperinsulinemic/euglycemic clamp conditions, is enhanced in congenital mouse models of GHD (GHRH^−/−^; [110]) and defects of GHR signaling (GHR-391; [111]) and impaired in the context of elevated endogenous GH, due to the congenital loss of hepatic IGF1 negative feedback (LID; [113,114]). In stark contrast, GH levels are shown to be positively associated with insulin-mediated HGP in mouse models with alterations in endogenous GH production that occurs after birth (AOiGHD and HiGH; [112]). These discrepancies may be related to the developmental stage at the onset of GH modification and the severity of the GH alteration that could impact the ultimate actions of GH on WAT lipolysis, indirectly mediating hepatic insulin sensitivity. This is supported by a study showing mice with the congenital hepatocyte-specific loss of JAK2 (JAK2L; leading to the loss of GHR signaling and elevated GH levels due to the loss of IGF1 negative feedback) have impaired suppression of HGP under hyperinsulinemic/euglycemic clamp conditions, and this is corrected by blocking the actions of GH on WAT via the concomitant loss of JAK2 in adipose tissue [115]. These results suggest GH-mediated WAT lipolysis increases NEFA availability to the liver to antagonize the actions of insulin, where lipolysis has been linked to hepatic insulin resistance [116,117]. However, in this same study [115], despite the fact that the concomitant loss of hepatocyte and adipose tissue GHR-signaling improves physiologic endpoints of hepatic insulin signaling (i.e., insulin-mediated suppression of HGP), this is not associated with alterations in the activation of canonical insulin-mediated downstream signals (pAKT and PRAS40). The impact of insulin on GH signaling (but not vice versa) was explored using a rat hepatoma cell line, H4IIE, and found to be positive with short-term treatment, but negative after long-term exposure [118]. In addition, only a limited number of studies combining GH and insulin treatment in primary hepatocyte cultures have been conducted and show enhanced or no change in canonical insulin-mediated downstream signals [119,120]. Therefore, additional studies are required to understand how GH directly mediates hepatocyte responsiveness to insulin. Curiously, the overexpression of the long form of the PRLR (that can activate STAT5) in HepG2 cells, primary hepatocytes, and livers of mice, enhances hepatic insulin signaling in a STAT5-dependent manner [121]. Based on these diverse observations, additional studies are required to understand the ultimate impact of GH/GHR on hepatocyte insulin/INSR and post-GHR signals involved.

Taken together, the complex interdependency between GH, IGF1, and insulin represents a challenge when conducting studies to determine what actions of GH on hepatocyte metabolism are direct or indirect, since GH regulates the production and action of both IGF1 and insulin, which in turn impacts GH production and function.

## 4. Circulating GH Levels Are Dependent on Age, Time of Day, and Nutritional Status

In humans, circulating GH levels are highest during the prepubertal period and then decline ~10% per decade of life [122,123]. In humans, but not rodents, this is associated with a decline in circulating IGF1 levels [123]. In normal lean humans, mean GH levels are low during the day in the context of meal-induced increases in nutrients and insulin. A diurnal pattern of GH in rodents (which are nocturnal) is not clearly established [23,124,125]. If normal patterns of food intake and sleep are maintained, total circulating IGF1 levels remain relatively constant throughout a 24 h period; however, in humans, it was shown that nocturnal-free IGF1 is decreased, related to a rise in IGFBP1 [126]. As previously discussed, under more extreme catabolic conditions (prolonged fasting or diabetes type I), circulating GH levels are elevated, while IGF1 is reduced [127]. In mouse models, the rise in GH in catabolic states has been attributed to a rise in the gastrointestinal hormone, ghrelin, which signals through the GH secretagogue receptor (GHSR) on pituitary somatotropes to enhance GH secretion [128,129]. The reduction in circulating IGF1 in catabolic states has been linked to the development of hepatic GH resistance (see below). When nutrients are in excess (short-term overeating, obesity, and diabetes type II), circulating GH levels are suppressed [26,32,33,34,130]. In some obese individuals, GH levels are as low as those with primary GHD [32,131] with variable effects on circulating IGF1 [132], which has been associated with the level of adiposity (a source of IGF1) and liver health [133,134,135]. However, with weight loss, GH levels rise to age-appropriate levels [136,137]. As reviewed above, alterations in insulin and IGF1 can contribute to nutritional-dependent changes in GH. In addition, many other factors contribute to the metabolic control of GH secretion including glucose, NEFA, glucocorticoids, and leptin, acting at the level of hypothalamic neurons or directly on pituitary somatotropes [138,139].

## 5. Role of GH in Regulating Hepatocyte Metabolism in Catabolic States

As discussed in the preceding sections, in humans, the fasting-induced rise in GH promotes WAT lipolysis through STAT5-independent mechanisms, leading to a rise in circulating glycerol and NEFA [17]. In mice, a rise in GH is also required to maximize WAT lipolysis in response to prolonged food restriction [37]. The mobilization of fat stores supplies glycerol that is used as a substrate for hepatic gluconeogenesis, as well as NEFA that is oxidized to provide a carbon substrate for ketogenesis (acetyl-CoA) and mitochondrial bioenergetics (ATP and NADH) to facilitate gluconeogenesis. Hepatic fatty acid oxidation is increased in fasting; however, a transient steatosis develops in mouse models, attributed to an imbalance between the influx/esterification and the utilization of NEFA from lipolysis [140], due in part to the induction of torpor, thereby reducing the amount of energy required for thermogenesis. The evidence is clear that the fasting-induced rise in GH can indirectly support hepatic gluconeogenesis, fatty acid oxidation, and ketogenesis by enhancing the supply of necessary substrates, as illustrated in Figure 2 (left panel). In addition, evidence showing GHR in the brain modulates metabolic function relevant to food restriction and exercise is emerging [141,142,143]. However, our understanding of the direct role GH plays in regulating hepatocyte metabolism under fasting conditions remains to be clarified.

### 5.1. Fasting-Induced Hepatic GH Resistance

As previously noted, despite the rise in GH, circulating IGF1 levels are reduced in fasting, indicating the liver becomes GH-resistant [127]. In rodent models, the fasting-induced GH resistance is typically studied in rats or mice after 24–48 h of fasting and is associated with an impairment in the GH-mediated phosphorylation of STAT5 [144,145]. Multiple mechanisms may contribute to the development of fasting-induced, hepatic GH resistance. Insulin receptor signaling via IRS1/IRS2 is required to maintain the expression of both GHR and IGF1 [99]. Therefore, low circulating insulin levels in the fasted state could contribute to a reduction of hepatic GHR protein and circulating IGF1 with prolonged fasting [99]. The fasting-induced reduction in hepatic GHR has also been attributed to the enhanced production of leptin receptor overlapping transcript (LEPROT) that is shown to bind to the GHR, promoting its internalization and degradation [146]. In addition, sirtuin 1 is enhanced in the fasted liver and is shown to induce GH resistance by deacetylating STAT5, thereby impairing GH-mediated STAT5 phosphorylation [147]. Sirtuin 1 also enhances the hepatic expression of FGF21, shown in an overexpression model to reduce GH-mediated STAT5 signaling and IGF1 expression [148]. Sirtuin 1 is also required to activate PPARα, a nuclear transcription factor critical for fasting-induced mitochondrial β (fatty acid)—oxidation [149]. Of note, PPARα is shown to block pSTAT5-mediated transcription in an in vitro system [150]. Studies conducted using ad libitum fed, whole body PPARα knockout mice showed bovine GH infusion (7d) decreases hepatic triglyceride content and increases hepatic triglyceride clearance in a PPARα-independent fashion [151]; however, it remains to be determined, if PPARα is required for fasting-induced reduction in GHR-signaling. To date, studies examining hepatic GH resistance have focused on the GH/GHR-mediated activation of STAT5; however, the GHR also signals through other pathways [3,4], which may be differentially impacted by fasting.

Taken together, when considering the direct effects of GH on mediating hepatic metabolism in fasted rodent models, it is important to consider the duration and severity of food deprivation, where the direct actions of GH on hepatocyte metabolism may be more evident in the context of the diurnal sleep (fast)/wake (feed) cycle and minimized with long-term caloric restriction.

### 5.2. Does GH Directly Regulate Hepatocyte Glycogenolysis, Gluconeogenesis, Ketogenesis, or Fatty Acid Oxidation?

#### 5.2.1. HGP (Glycogenolysis and Gluconeogenesis)

With short-term fasting, for example occurring with normal sleep wake cycles, a primary source of hepatic glucose is derived from the breakdown of glycogen (glycogenolysis), with gluconeogenesis playing a more critical role when glycogen stores are depleted (collectively referred to as HGP). In healthy human subjects, the blockade of GH production using a GHRH antagonist, during normal sleep/wake cycles has limited impact on lipolysis or HGP but reduces both lipolysis and HGP after two days of fasting [152]. Importantly, GH-mediated increase in HGP is blunted but not eliminated, when WAT lipolysis is blocked by acipimox, suggesting in humans GH indirectly and directly regulates HGP [153]. In fasted subjects, the infusion of GH during a hyperinsulinemic/euglycemic clamp is shown to enhance HGP by sustaining glycogenolysis, without altering gluconeogenesis [154,155]. In acromegalics (a state of chronic GH excess), gluconeogenesis is enhanced in some, but not all, studies [156,157,158].

Studies using mouse models of hepatocyte-specific knockout of the GHR also support both the direct and indirect actions of GH in promoting HGP. Mice with congenital liver-specific GHR knockout (LiGHRKO), a model of elevated GH, due to the loss of IGF1 negative feedback, exhibit insulin resistance, WAT lipolysis, and enhanced glucose production after pyruvate challenge (pyruvate tolerance test (PTT)), indicative of increased gluconeogenesis [159]. However, when LiGHRKO mice are crossbred with mice that express a heterologous IGF1 transgene (HIT), which reduces circulating GH, insulin, and NEFA levels, the glucose response to the PTT is actually less than that observed in GHR-intact controls, suggesting the hepatocyte GHR is required to sustain gluconeogenesis [159]. In addition, mice with short-term (seven-day) adult-onset hepatocyte-specific GHR knockdown (aHepGHRkd) show the reduced hepatic expression of the gluconeogenic genes (glucose-6-phosphatase catalytic subunit 1 (*G6pc*) and phosphoenolpyruvate carboxykinase 1 (*Pck1*); [59]) and impaired glucose production in response to PTT (unpublished data), without the evidence of altered WAT lipolysis or enhanced hepatic (canonical) insulin signaling [59]. In addition, a more recent report overexpressed the mGHR in hepatocytes using an AAV8-Albumin-mGHR and reported an increase in gluconeogenesis as assessed by the PTT [160].

Although the PTT is a simple method to assess hepatic gluconeogenesis, it should be noted that alterations in pyruvate uptake and competition with glucose in oxidative processes in extrahepatic tissues can confound the interpretation [161]. Nonetheless, these in vivo studies coupled with in vitro studies conducted in the absence of insulin support a direct role of GH/GHR in enhancing hepatic gluconeogenesis. Specifically, ovine GH increases endpoints related to gluconeogenesis in sheep hepatocytes [162]. In a mouse hepatocyte cell line that expresses the GHR (AML-12; [163]) or in primary rat hepatocytes, hGH enhances *Pck1* and *G6pc* expression (mRNA and protein), through a JAK2/STAT5-dependent mechanism [164]. In addition, hGH enhances pyruvate dehydrogenase kinase (PDK4), reducing pyruvate dehydrogenase complex (PDC) phosphorylation (activation) and acetyl-CoA levels in a STAT5-dependent manner, which would serve to preserve pyruvate for gluconeogenesis [165].

It should be noted that despite the above reports showing a positive relationship between GH/GHR and hepatic gluconeogenesis, a rise in GH is not required to sustain glucose levels after severe chronic caloric restriction (a model of WAT depletion) in either AOiGHD mice [37] or mice with somatotrope-selective GHSR deletion [129]. However, an intact hepatocyte GHR is required to sustain glucose after severe chronic caloric restriction, where mice with congenital hepatocyte-specific GHR knockout become severely hypoglycemic, associated with a reduction in hepatic autophagy, perhaps leading to a reduction in ATP necessary to sustain gluconeogenesis [166], with amino acids supplying the necessary substrate in this context.

#### 5.2.2. Ketogenesis

Consistent with GH-mediated lipolysis supplying NEFA for hepatic ketogenesis, in humans, GH levels are positively associated with circulating ketones [167,168], where the GH receptor antagonist, pegvisomant, reduces the fasting-induced rise of NEFA and ketones [169]. It was suggested that GH may work directly on the hepatocyte to control ketogenesis. This is based on a study in prediabetic/diabetic patients where treatment with the sodium-glucose cotransporter 2 (SGLT2) inhibitor, empagliflozin, leads to an increase in β-hydroxybutyrate levels associated with a decrease in insulin/glucagon ratio, as well as glucose and NEFA levels, when studied after a meal where the primary source of NEFA is from the diet. When patients are co-treated with empagliflozin and pegvisomant, the rise in β-hydroxybutyrate is blocked, without significantly impacting the other endpoints [170]. In addition, hepatic β-hydroxybutyrate levels are reduced in aHepGHRkd mice in the post-absorptive state, independent of changes in glucose and NEFA levels, or the evidence of enhanced hepatic insulin signaling [59]. However, in the same study that showed oGH treatment enhances gluconeogenesis in sheep hepatocyte cultures, there was no evidence that GH increases fatty acid flux to ketones [162]. It may be speculated that this is due to GH-mediated increase in PDK4 that reduces the PDC conversion of pyruvate to acetyl-CoA, the substrate for ketogenesis [165].

#### 5.2.3. Fatty Acid Oxidation

As indicated above, GH-mediated lipolysis generates NEFA, which serves as a substrate for mitochondrial β (fatty acid) oxidation and ATP generation. In acromegalic subjects, hepatic lipid content is low and is associated with an increase in hepatic ATP synthesis rate as assessed by ^31^P/^1^H-7-T MR spectroscopy [171]. In addition, aHepGHRkd mice under basal conditions suppress the hepatic expression of carnitine palmitoyltransferase 1A (*Cpt1a*), independent of changes NEFA levels, or enhanced hepatic insulin signaling [59]. Whether direct actions of GH on hepatocytes contributes to the positive association between GH and endpoints of hepatic β-oxidation remains to be determined. Short-term in vivo experiments in dogs [172] and in vitro experiments using rat liver slices [173] showed substrate (fatty acid) availability is positively associated with β-oxidation, but GH does not further promote β-oxidation. However, hGH is shown to enhance β-oxidation in human fibroblasts [174]. Of note, GH may impact hepatic β-oxidation through the STAT5-mediated suppression of BCL6 [175], a transcription factor recently shown to be a repressor of the β-oxidative program [176]. In addition to the possible regulation of ATP generation through β-oxidation, as previously discussed, hepatocyte GHR signaling appears to be required to sustain hepatic autophagy as a source of energy, when WAT stores are depleted [166].

## 6. Role of GH in Regulating Hepatocyte Metabolism in States of Nutrient Excess

### 6.1. Associations between Alterations in GH Production and Signaling and Metabolic Disease

Excess nutrient intake and reduced activity leads to obesity, development of insulin resistance, diabetes type II, and increased accumulation of hepatic triglycerides (TG), referred to as steatosis. Excess fat accumulation can progress to hepatocyte ballooning and inflammation, with or without fibrosis, referred to as non-alcoholic steatohepatitis (NASH), which increases the risk of cirrhosis and hepatocellular carcinoma [177,178]. The term non-alcoholic fatty liver disease (NAFLD) is used to encompass the spectrum from simple steatosis to NASH. There are multiple mechanisms that may drive excess fat accumulation in the liver, including the following: (1) enhanced de novo lipogenesis (DNL), primarily from excess dietary carbohydrates, (2) an increase in the uptake and esterification of NEFA from dietary fat or from WAT lipolysis due to inflammation and insulin resistance, (3) impairment in hepatic VLDL-TG release from the liver, and (4) reduced hepatic fatty acid utilization (β-oxidation). Although insulin can drive each of these processes, in the context of obesity-associated NAFLD, it was recently suggested that both the insulin-mediated suppression of HGP and the induction of lipogenesis are impaired [179], but these results remained to be confirm by others. However, evidence indicating enhanced DNL is a major contributor to NAFLD progression is accumulating, likely fueled by excess dietary carbohydrate intake, as well as an increase in substrate flux to the liver, as a result of hepatic and systemic insulin resistance [179,180,181]. In this context, circulating GH levels are reduced [26,32,33,34,131], and the question remains, if the reduction in GH directly contributes to the progression of steatosis and NASH (Figure 2, right panel).

Steatosis in humans has been associated with low circulating GH levels [182,183] and IGF1 levels [184,185,186,187,188,189,190,191,192,193], as well as inactivating mutations in the GHR [194]. In addition, GWAS data collected from patients with NAFLD revealed single nucleotide polymorphisms (SNPs) in 18 different genes in the GH signaling pathway, including *Gh*, *Ghr*, *Jak2*, and *Stat5b,* are associated with steatosis [195]. This association suggests individuals with modifications in the GH-signaling pathway may be more susceptible to NAFLD. In addition, therapies that increase GH reduce steatosis in young men with abdominal obesity [196], patients with HIV lipodystrophy [197,198], and patients with primary GHD [182,199,200]. In fact, a reduction in hepatic lipid content observed in GH-treated HIV patients is associated with a decrease in hepatic fractional DNL [198]. In addition, the blockade of GH signaling using the GHR antagonist, pegvisomant, increases hepatic fat content in acromegalic patients [201]. In line with these clinical findings, mice that express a GHR antagonist develop steatosis [202]. Additionally, mice with diet-induced obesity/steatosis exhibit reduced circulating GH levels [22,33], where GH treatment or elevation in endogenous GH [203,204,205] or GH-signaling due to the loss of SOCS2 [206] decreases diet-induced steatosis. Although there is a clear negative association between GH production and signaling and hepatic lipid content, the question remains if these actions are direct or indirect.

### 6.2. Indirect vs. Direct Effects of GH on Steatosis

Consistent with the negative association between GH and steatosis, mouse models with the congenital, liver-specific knockout of the GHR or its downstream effectors (JAK2 or STAT5) as adults exhibit steatosis, glucose intolerance, insulin resistance, and WAT lipolysis [207,208,209,210,211]. In these congenital knockout models, it is believed that fatty liver develops due to the indirect action of GH. As discussed previously, the experimental loss of the hepatic GHR/JAK2/STAT5 signal cascade dramatically reduces circulating IGF1 levels, leading to a rise in circulating GH levels and thereby reducing systemic insulin sensitivity, leading to hyperinsulinemia, hyperglycemia, and WAT lipolysis. These changes in systemic metabolic function shift the flux of glucose, glycerol, and NEFA to the liver, thus providing substrates for TG formation. The important role of the reciprocal shift in IGF1/GH and subsequent GH-mediated WAT lipolysis to the development of steatosis in these model systems is based on studies in mice with the loss of hepatocyte GHR signaling, due to the congenital liver-specific knockout of JAK2 (JAK2L). When JAK2L mice are crossbred to a GH-deficient mouse model (lit/lit; [211]) or crossbred to mice with the adipose-tissue knockout of JAK2 [115,212], steatosis is dramatically reduced.

Hepatocyte GHR signaling appears to also play a direct role in controlling hepatic fat accumulation and preventing liver injury in mice, even in the context of standard chow-diets. As previously discussed, the elevated GH levels and impaired systemic metabolic function observed in mice with the congenital LiGHRKO are normalized by crossbreeding to mice expressing a liver-specific heterologous IGF1 transgene (HIT). Although steatosis and associated hepatic inflammation and oxidative stress are reduced in this model, it is not eliminated [159]. In addition, mice with the adult-onset loss of hepatocyte GHR signaling (aHepGHRkd) exhibit a reciprocal shift in circulating IGF1 and GH levels, which is similar to congenital knockout models. However, unlike mice with the congenital loss of hepatocyte GHR signaling, aHepGHRkd mice only develop mild insulin resistance, associated with an increase in insulin levels, without alterations in glucose tolerance, WAT lipolysis, or hepatic VLDL-TG release [59,213]. In male aHepGHRkd mice, steatosis rapidly develops and persists up to one year of age, with evidence of hepatocyte ballooning, inflammation, and mild fibrosis, indicative of early signs of NASH [62,213]. Of note, female aHepGHRkd mice are protected from steatosis and liver injury [62], similar to that reported in female mice with congenital LiGHRKO [209]. To further isolate the role of hepatocyte GHR signaling from changes in IGF1, aHepGHRkd mice are treated with AAV-TBGp-ratIGF1 that restores circulating IGF1 1evels, sufficient to normalize GH and insulin, but steatosis and liver injury persist [62], further supporting a direct role of hepatocyte GHR signaling in suppressing excess hepatic fat accumulation.

### 6.3. Potential Mechanisms by Which GH Directly Regulates Hepatocyte Lipid Accumulation

Evidence collected so far has indicated steatosis that develops after aHepGHRkd cannot be clearly attributed to the induction of hepatic PPARγ activity (a lipogenic transcription factor), enhanced fatty acid uptake, or impaired VLDL-TG release [59,60,62]. However, the steatosis that rapidly develops in the aHepGHRkd model is associated with enhanced DNL in the post-absorptive state, as assessed by the deuterated water labeling of newly formed fatty acids [59]. In addition, as assessed by GC/MS, alterations in fatty acid composition indicative of enhanced DNL persist with age in aHepGHRkd mice, as well as LiGHRKO mice, even after the reconstitution of IGF1 that normalizes circulating GH and systemic metabolic function [62,159,213]. In livers of aHepGHRkd mice, the upregulation of DNL is associated with endpoints of enhanced glycolysis, including increased glucokinase expression (*Gck* mRNA and cytoplasmic protein levels), as well as an increase in fructose 2,6 bisphosphate, the most potent activator of glycolysis and inhibitor of gluconeogenesis [59,62,213]. When nutrients are adequate, downstream products of GCK supply substrates for DNL and activate the lipogenic transcription factor, carbohydrate response element-binding protein (ChREBP; [214,215,216]). These results suggest GH/GHR directly suppresses glycolysis-driven DNL. Therefore, the reduction in GH levels observed in obesity may directly contribute to the high DNL observed in patients with NAFLD [179,180,181].

Many questions remain regarding the exact mechanism(s) by which the loss of hepatocyte GHR signaling brings about these changes. Several possibilities remain to be explored. (1) Although no clear evidence has been published demonstrating a direct inhibitory action of GH/GHR on canonical insulin/INSR signaling, it remains to be tested if the hepatocyte loss of the GHR might improve insulin signaling and drive DNL. (2) Alternatively, hepatocyte GH/GHR signaling through STAT5-dependent or -independent mechanisms may work in parallel to insulin, providing a rheostat to modulate flow of substrates between glucose production and glucose storage. For example, as reviewed above, there is evidence that GH/GHR/STAT5 may directly stimulate gluconeogenesis [162,163,164], and therefore loss of this stimulation could shift a substrate towards DNL, as observed in mice with congenital hepatocyte-specific Pck1 knockout [217]. Certainly, additional studies are required to tease apart the direct actions of GH from insulin in regulating these processes. Future studies could include studying aHepGHRkd mice (without or with the correction of select downstream signals) under hyperinsulinemic/euglycemic clamps conditions, using stable isotope tracers to track the flux of carbohydrates and lipids through these metabolic pathways.

## 7. Past and Future

Since the first biochemical characterization of GH in 1973 [218], the scientific community has come a long way towards understanding how GH regulates metabolic function. However, the recent development of new animal models to modulate GH signaling in a tissue- and time-dependent fashion, coupled with sensitive analytical and imaging techniques to trace metabolic fluxes in both humans and animals, will dramatically accelerate our progress toward understanding the direct and indirect actions of GH on hepatocyte carbohydrate and lipid metabolism.

## Figures and Tables

**Figure 1 cells-10-02532-f001:**
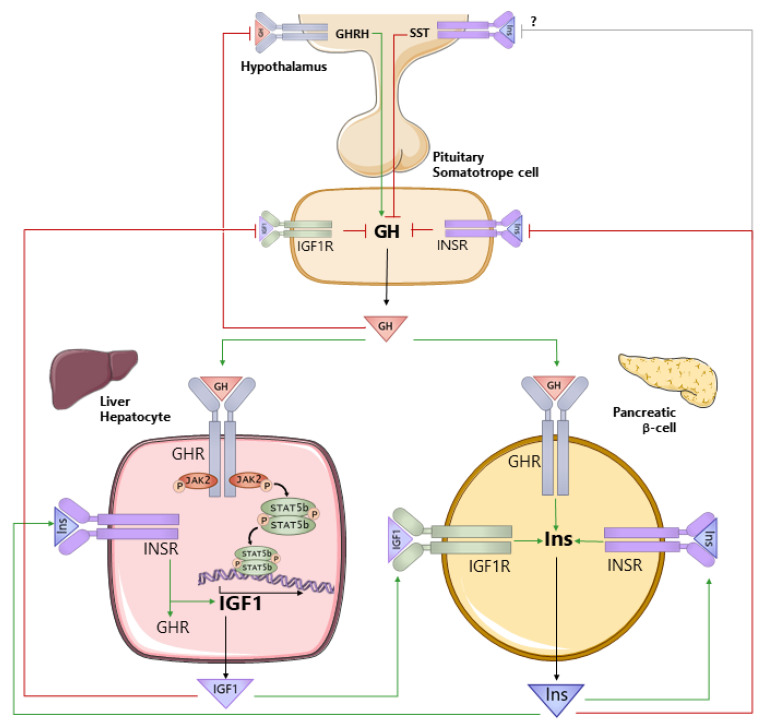
Interrelationship between growth hormone (GH), insulin-like growth factor 1 (IGF1), and insulin (Ins). GH is produced by somatotropes within the anterior pituitary gland, under the primary control of the hypothalamic neuropeptides, growth hormone-releasing hormone (GHRH), and somatostatin (SST). GH is released into the general circulation and acts through the GH receptor (GHR) to stimulate the production of IGF1, where the liver (hepatocyte) is the primary source of circulating IGF1. GH negatively feeds back at the level of the hypothalamus to control its own production. In addition, IGF1 negatively feeds back at the level of the somatotrope to suppress GH synthesis and release. Both GH and IGF1 support the production of insulin from pancreatic β-cells, and insulin acts through its own receptor (INSR) to support its own production. Insulin also acts directly on the hepatocyte to maintain GHR and IGF1 production. At the level of the pituitary somatotrope, insulin suppresses GH secretion. Green lines (ending >) depict positive actions, and red lines (ending |) depict negative actions.

**Figure 2 cells-10-02532-f002:**
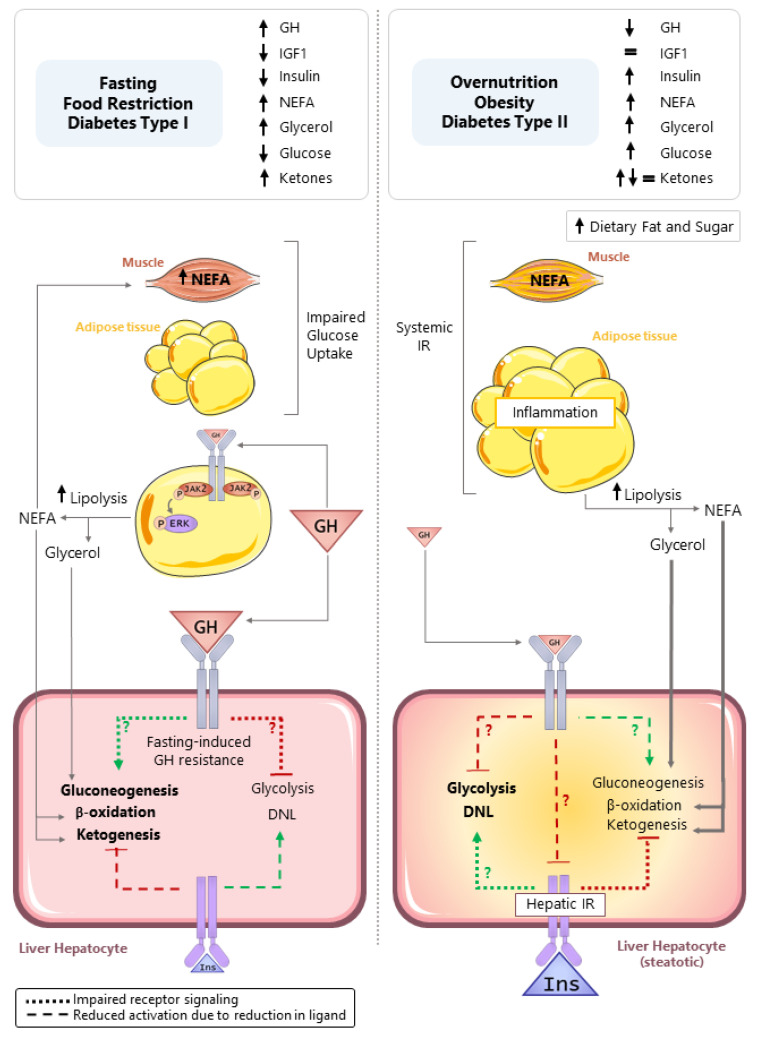
Direct and indirect actions of GH in controlling hepatocyte carbohydrate and lipid metabolism in response to nutrient deprivation (fasting, food restriction, diabetes type (I), and nutrient excess (overnutrition, obesity, and diabetes type (II)). Left panel: In the context of nutrient deprivation, circulating insulin and IGF1 levels are reduced, while the GH level is increased. GH promotes white adipose tissue lipolysis, leading to an increase in circulating non-esterified fatty acids (NEFA) and glycerol that serve as substrates for hepatic gluconeogenesis, fatty acid (β) oxidation, and ketogenesis. The NEFA released from lipolysis impairs glucose uptake in the skeletal muscle, due to substrate competition. There is suggestive evidence that GH directly stimulates gluconeogenesis and plays a role in ATP generation to sustain gluconeogenesis. However, it should be noted that prolonged fasting leads to hepatic GH resistance. Therefore, these direct actions of GH on gluconeogenesis may be more relevant in normal diurnal cycles of feeding and fasting. Right panel: In the context of nutrient excess, obesity develops and can lead to skeletal muscle and WAT insulin resistance, linked to inflammation. This is associated with the development of hepatic insulin resistance and steatosis. In this context, circulating GH levels are reduced. Evidence collected from mouse models with the loss of hepatocyte GHR indicates the GHR is required to suppress steatosis by inhibiting glycolysis driven de novo lipogenesis (DNL). The exact mechanisms driving enhanced DNL after hepatocyte-specific loss of GH signaling remains to be determined and could include the following: (1) derepression of hepatic insulin signaling; (2) a reduction in gluconeogenesis shifting substrates toward DNL; and/or (3) regulating, yet to be determined, endpoints that alter glycolysis-driven DNL. Green lines (ending >) depict positive actions, and red lines (ending |) depict negative actions. Dotted lines (…) indicate reduced signaling due to impaired receptor signaling, while dashed lines (---) indicate reduced pathway activation due to a reduction in ligand.

## Data Availability

This review does not contain original data.

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
