# Peer review of "Towards Understanding the Direct and Indirect Actions of Growth Hormone in Controlling Hepatocyte Carbohydrate and Lipid Metabolism"

_cells, 2021, doi:10.3390/cells10102532_

Round 1

Reviewer 1 Report

The review article by Vazquez-Borrego is excellent in many ways. It goes well beyond most reviews in terms of representing the literature in a balanced manner and addressing many of the challenges associated with studying growth hormone action and hepatic metabolism. The consideration of studying hepatocytes in culture in not typically represented in such a transparent manner. The figures are novel and substantially add to the text. One consideration might be to compare GH as a modulator of lipolysis to other activators and inhibitors of lipolysis. For example, the actions of the repressive effects of insulin that is acute and activation by glucocorticoids or TNF that is chronic but much greater than the effects of GH on lipolysis.

Author Response

Reviewer 1 - One consideration might be to compare GH as a modulator of lipolysis to other activators and inhibitors of lipolysis. For example, the actions of the repressive effects of insulin that is acute and activation by glucocorticoids or TNF that is chronic but much greater than the effects of GH on lipolysis.

Response - This is a very interesting comparison and certainly important, since GH is not acting in isolation.  However, in this review, the authors have attempted to focus discussion on the known actions of GH on adipocyte function.

Reviewer 2 Report

The author has thoroughly discussed the direct and indirect actions of GH to control liver carbohydrate and lipid metabolism in different conditions. Besides, they provide sufficient content to discuss the model systems used and areas that require further investigation. However, some minor issues need to be addressed to help the readers better understand the content. 

  1. The author had mentioned different concepts in section 2, providing the subtitle will help the content clearer and easier to be understood. The same issue can also be found in section 6.
  2. The order in section 5 is confusing. Why some of the italic types is with or without numbering? 

Author Response

Reviewer 2. 1 - The author had mentioned different concepts in section 2, providing the subtitle will help the content clearer and easier to be understood. The same issue can also be found in section 6.

Response - This is an excellent suggestion.  Subtitles have been added where appropriate within sections 2-6.

Reviewer 2.2 - The order in section 5 is confusing. Why some of the italic types is with or without numbering? 

Response - Additional subheadings were added to clarify the designation of specific paragraphs in this section.

Reviewer 3 Report

This well-written and very informative review provides in-depth discussion of the effects of GH on liver carbohydrate and lipid metabolism.  Based on their own work and data from other labs, the authors address the very complex issues of direct vs indirect effects, interactions between GH, IGF-1 and insulin signaling, acute vs chronic effects and insulin resistance in the liver as compared to other tissues.

A few minor issues need attention and the authors may wish to consider expansion or clarification of some parts of the text, as listed below.

Line 12  and elsewhere in the manuscript:   GHR is usually classified as Class I cytokine receptor rather than a tyrosine kinase receptor (even though several aspects of GH signaling and its interaction with Jak2 seem to fit each of these two categories).  Is this a newer classification system?  Should this perhaps be explained?

Line 100  Ref 30 refers to 17AE2 which is not a major gonadal steroid

11- 113  It might help the reader to mention that these differences are in addition to the relationship of BMR to body weight

116 these rodents or laboratory rodents  (rather than “rodents” )

185  dominate,  are dominant ?

199/200  numbered citations ?

330 perhaps thermogenesis should be mentioned here

Section starting on line 340:  The authors may want to include discussion of inhibitory (serine) phosphorylation of IRS in insulin resistance or at least mention that this mechanism may not play a major  role in the liver

381 typo; determined

422  treatment with…

Author Response

Reviewer 3.1 - Line 12  and elsewhere in the manuscript:   GHR is usually classified as Class I cytokine receptor rather than a tyrosine kinase receptor (even though several aspects of GH signaling and its interaction with Jak2 seem to fit each of these two categories).  Is this a newer classification system?  Should this perhaps be explained?

Response - This has been corrected.

Reviewer 3.2 -Line 100  Ref 30 refers to 17AE2 which is not a major gonadal steroid

Response - This has been qualified by using the broader term “estrogens”

Reviewer 3.3 -11- 113  It might help the reader to mention that these differences are in addition to the relationship of BMR to body weight.

Response - This concept is now incorporated.

Reviewer 3.4 -116 these rodents or laboratory rodents  (rather than “rodents” )

Response - “laboratory” has been added.

Reviewer 3.5 - 185  dominate,  are dominant ?

Response - This has been corrected.

Reviewer 3.6 -199/200  numbered citations ?

Response - The databases used are now shown in text as reference numbers and the URLs are provided in the bibliography.

Reviewer 3.7 -330 perhaps thermogenesis should be mentioned here

Response - Related fasting induced steatosis in mice and thermogenesis, the following was added.

“… due in part to induction of torpor, thereby reducing the amount of energy required for thermogenesis.”

Reviewer 3.8 -Section starting on line 340:  The authors may want to include discussion of inhibitory (serine) phosphorylation of IRS in insulin resistance or at least mention that this mechanism may not play a major role in the liver.

Response - Although this is a valid point, in this section we attempted to focus on the mechanisms related to fasting induced hepatic GH resistance.

Reviewer 3.9 - 381 typo; determined

Response - corrected

Reviewer 3.10 - 422  treatment with…

Response - corrected